# ATP-powered molecular recognition to engineer transient multivalency and self-sorting 4D hierarchical systems

Jie Deng [1,2,3,4,5] & Andreas Walther [1,2,3,4,5✉]

Biological systems organize multiple hierarchical structures in parallel, and create dynamic assemblies and functions by energy dissipation. In contrast, emerging artificial non-equilibrium self-assembling systems have remained relatively simplistic concerning hierarchical design, and non-equilibrium multi-component systems are uncharted territory. Here we report a modular DNA toolbox allowing to program transient non-equilibrium multi-component systems across hierarchical length scales by introducing chemically fueled molecular recognition orchestrated by reaction networks of concurrent ATP-powered ligation and cleavage of freely programmable DNA building blocks. Going across hierarchical levels, we demonstrate transient side-chain functionalized nucleic acid polymers, and further introduce the concept of transient cooperative multivalency as a key to bridge length scales to pioneer fuel-driven encapsulation, self-assembly of colloids, and non-equilibrium transient narcissistic colloidal self-sorting on a systems level. The fully programmable and functiona-lizable DNA components pave the way to design chemically fueled 4D (3 space, 1 time) molecular multicomponent systems and autonomous materials.

[1] A3BMS Lab, Institute for Macromolecular Chemistry, University of Freiburg, Stefan-Meier-Straße 31, 79104 Freiburg, Germany. [2] DFG Cluster of Excellence "Living, Adaptive and Energy-Autonomous Materials Systems" (livMatS), 79110 Freiburg, Germany. [3] Freiburg Materials Research Center, University of Freiburg, Stefan-Meier-Straße 21, 79104 Freiburg, Germany. [4] Freiburg Center for Interactive Materials and Bioinspired Technologies (FIT), University of Freiburg, Georges-Köhler-Allee 105, 79110 Freiburg, Germany. [5] Freiburg Institute for Advanced Studies (FRIAS), University of Freiburg, Albertstraße 19, 79104 Freiburg, Germany. ✉email: Andreas.walther@makro.uni-freiburg.de

Biological self-organizing systems are characterized by molecularly programmed hierarchical structures, that are compartmentalized and self-sorted, and which run on chemical fuels and are orchestrated through reaction networks to provide capacity for active, adaptive and autonomous non-equilibrium behavior[1–3]. On a systems level this gives rise to cellular transport[4], cell proliferation[5], mobility[6], and signal transduction[7]. Individual aspects of the underlying biological principles have been successfully mimicked in synthetic systems[8–10]. Equilibrium self-assembly from diverse building blocks allows complex hierarchical structures in solution and bulk, and encoding switchability leads to responsive materials such as in structural, optical or bioactive materials for actuators and soft robotics[3,11–14]. A recent extension in equilibrium self-assembly was the move towards self-sorting multicomponent systems, most notably from the peptide and colloidal self-assembly fields[15,16].

Inspired by nature, there is an emergent pursuit to develop man-made non-equilibrium systems showing active and life-like behaviors[17–21], such as adaptability and autonomous operation. Synthetic systems requiring continuous energy influx to be operational bring new opportunities to functional materials and have been reported for molecular structure switching[22], supramolecular polymerization[23,24], nanostructures and reactors[25–28], and transient hydrogels and photonic materials[29,30]. However, such fuel-driven structures have not attained similar control over hierarchical architectures, let alone multicomponent self-sorting. Hence, a critical cross-fertilization of concepts is needed to implement higher biology-inspired systems complexity. Two critical bottlenecks that need to be addressed are (i) the development of fuel-driven molecular recognition to allow for parallel selectivity in multicomponent systems and (ii) finding an ability how molecular-scale fuel-driven process transduce through hierarchical layers.

DNA is highly promising for designing autonomous systems to address the aforementioned challenges due to its high programmability[31]. However, so far most examples of DNA-based molecular motors[32], walkers[33], or transport systems[34–36] are thermodynamically down-hill, and only few studies focused on fuel-driven non-equilibrium self-assembled DNA structures using temporary up-hill process, which, however, is essential for programming transient autonomous lifecycles and adaptive steady-states[37]. Most notable examples include enzyme-assisted

transcription/degradation machineries that allow transient structures[38,39] or other reaction networks[40]. We recently demonstrated ATP-fueled dynamization of a covalent DNA bond via an enzymatic reaction network (ERN) of concurrent ligation and cleavage, and reported details on engineering the adaptiveness of the dynamics, pathway complexity, and light-modulated behaviors on a polymer level[41–43]. However, fully programmable recognition and sequence-controlled structures remain disallowed in this system due to the sequence-specific cleavage by BamHI (a Class I endonuclease). In stark contrast, class IIS endonucleases show prominent programmability regarding the ligation/cleavage sticky ends. Using such class IIS endonucleases, programmable DNA automatons to solve computational problems[44], and Golden Gate cloning for constructing circular plasmids were developed[45]. Yet, such class IIS endonucleases have not been put in the context of ATP-limited non-equilibrium self-assemblies with spatiotemporal structural complexity and functions translating across hierarchies in multicomponent systems.

Herein, we report a modular DNA toolbox that is able to program transient, flux-like structures (i.e. constant bond-shuffling) based on an ATP-driven programmable molecular recognition site. The excellent programmability of the system allows for the design of transient and dynamic steady-state sequence-defined functionalized nucleic acid polymer (SfNAP). By encoding spatial control over multivalent side group interactions, these transient SfNAPs allow to transduce the ATP-driven process into cooperative multivalency to transiently recognize colloidal partners to program fuel-driven encapsulation and self-assembly of colloid/DNA hybrids on the colloidal length scales. Due to the flexible programmability, multiple systems can be run in parallel to encode the spatiotemporal behavior of 4D multicomponent systems, pioneering transient narcissistic self-sorting in multiassembly colloidal systems.

## Results

**Design of the modular DNA toolbox to engineer ATP-fueled programmable molecular recognition.** We first establish the programmable toolbox for ATP-driven, transient molecular recognition by embedding a freely programmable DNA restriction site into an ERN of ATP-powered ligation using T4 DNA ligase and a specific class IIS restriction enzyme, BsaI (Fig. 1). Once a system containing the enzymes and the building blocks is

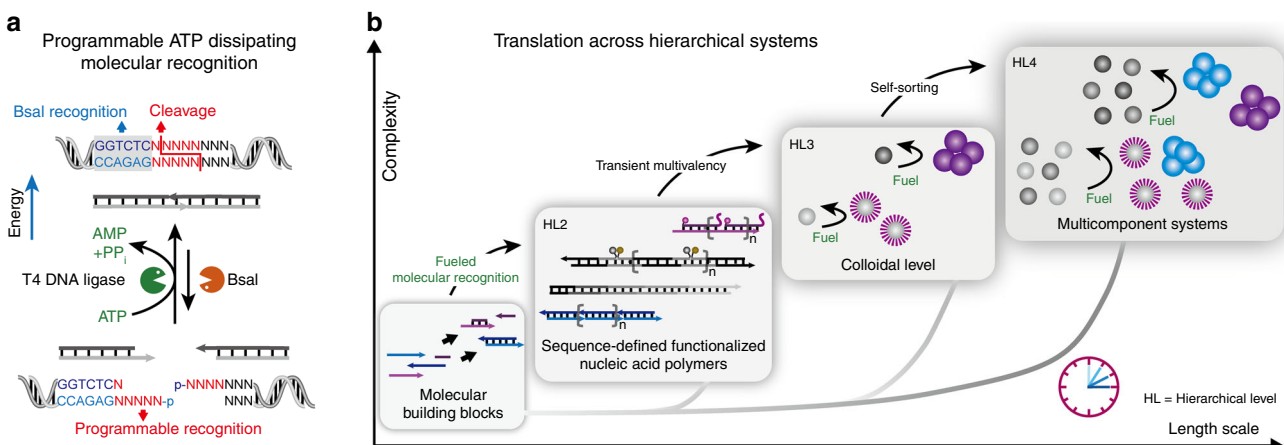

**Fig. 1 Hierarchical transduction of non-equilibrium structures. a** Illustration of the ATP-driven dynamic covalent DNA bond with fully programmable recognition, for which the lifetime is coupled to ATP fuel level, and in which the DySS properties are regulated by an enzymatic reaction network. **b** ATP-fueled modular DNA toolbox to engineer dissipative structures across 4D hierarchies, going from fuel-driven molecular recognition via sequence-defined functionalized nucleic acid polymers and encoded multivalency to self-sorting 4D multicomponent colloidal systems. HL indicates the hierarchical levels. HL2 is formed solely based on the covalent nature of the reaction cycle, while HL3 uses the covalent structure on HL2 to achieve higher level supramolecular structures by multivalency.

injected with ATP, the structures are driven into a non-equilibrium dynamic steady state (DySS) with an up-hill formation of a dynamic covalent phosphodiester bond by ligation at the complementary sticky ends on the DNA tiles and simultaneous hydrolysis by BsaI for monomer regeneration. DySS refers to an energy-driven out-of-equilibrium state, in which the frequency of ATP-driven bond formation and cutting is balanced, leading to a relatively stable, but dynamic ensemble average[41].

The reaction network fulfills the relevant criteria for dissipative non-equilibrium system: the structure formation is bound to an energy-fueled activation (ATP-dependent ligation), and the deactivation dissipates energy (cleavage of a covalent bond, $\Delta G = -5.3$ kcal/mol)[46]. Additionally, the activation and deactivation pathways are chemically independent, selective and kinetically tunable reactions, and the structure is completely reversible on a molecular level. The reaction network on this level is reminiscent of the non-equilibrium steady state phosphorylation/dephosphorylation networks in biology[47–49]. Strictly speaking the closed nature achieves a quasi steady state behavior with a behavior similar to an open steady state system with a perfect ATP regeneration system, because on short time scale the ATP/AMP concentration changes are negligibly small[50–55]. In contrast to biological driven systems, such as GTP-driven microtubules or ATP-driven actin, the structure itself does however not carry the catalytic function to hydrolyze the bound fuel. Here, the fuel is taken up and the kinetics of the reaction cycle are designed to allow for relaxation of the energy-rich state as commonly done in artificial non-equilibrium chemically-driven systems[23,24,30].

In contrast to our earlier work on a non-programmable restriction enzyme[41], the present design provides a step-change advance to freely design the sticky ends of the overhangs of the ligation/cleavage site because BsaI recognizes a sequence 5′-GGTCTC/GAGACC-3′, but cleaves the sequence downstream at 5′-GGTCTC(N1)/(N5)GAGACC-3′ without sequence specificity (Fig. 1a). This means that $4^4 = 256$ possibilities at the 4 nucleotide (nt) overlap junction can be programmed, opening considerable freedom in multicomponent systems design. This programmability allows to arrange tailor-made DNA tiles in precise sequences inside transient DySS DNA polymers achieving 100 % programmability regarding functional side group organization and enabling the formation of transient SfNAPs. Hence, ATP-fueled structures translate across four hierarchical levels from fueled and programmable molecular recognition to SfNAPs, colloidal hybrids and self-sorting systems using this modular DNA toolbox (Fig. 1b).

**ATP-fueled transient DySS DNA polymerization with programmable lifetimes.** For proof-of-concept, we first discuss the ATP-fueled transient DySS DNA polymerization for this particular ERN using α,ω-telechelic monomers (M1, Supplementary Table 1) that can be driven into an energy-fueled dynamic covalent polymer (Fig. 2a, b). The typical reactions contain 0.05 mM M1 and 0.46 Weiss Units (WU) $\mu L^{-1}$ T4 DNA ligase as well as 2.5 U $\mu L^{-1}$ BsaI at 37 °C. Agarose gel electrophoresis (AGE) of time-dependent aliquots visualizes the transient nature of the formed DySS polymers regarding their length distribution (Fig. 2c; Supplementary Fig. 1). A quantification of the AGE allows to calculate the average mass-weighted chain length in base pairs ($\overline{bp_w}$) from the mass-weighted distribution calibrated from standard gene rulers[41]. Gray scale profiles extracted from AGE at 1.0 mM ATP display the monomer band, M1, at ca. 40 base pairs (bp) before ATP injection (Fig. 2d). After ATP addition, the chains rapidly grow up to ca. 2500 bp within 10 min and reach a DySS plateau at ca. 10 kbp after 1 h. In the beginning, the ligation is favored as the system is rich in ATP fuel and because the

concentration of substrate is high, while restriction sites are still formed, limiting the speed of the restriction. Upon reaction progress the mutual feedback of both processes in the ERN lead to a balance in the DySS and to a constant frequency of ligation and cutting. Once the ATP cofactor is increasingly consumed, the ligation process seizes and the kinetic balance shifts towards the degradation. After two days, the DySS DNA polymer is degraded and the system decomposes to its original state. The $\overline{bp_w}$ is affected by the ATP concentration. At low ATP concentration (0.1 mM), the system hardly enters a DySS with balanced reaction rates as the concentration of ATP is only double of M1 (note that one junction requires two ATP molecules). Intermediate ATP concentrations (0.2 and 0.6 mM) promote the $\overline{bp_w}$ from 300 bp to around 1900 bp in a sustained DySS polymerization (Fig. 2e).

Overall, once we inject ATP the system undergoes an autonomous process to generate transient DNA structure, and, once the fuel has run out, the system degrades back to its original state, giving rise to an ATP-fueled autonomous system. The corresponding lifetimes of the autonomous systems, as defined by the point where the $\overline{bp_w}$ declines below half of the $\overline{bp_w}$ of the DySS plateau, can be easily tuned from hours to days by the ATP concentration (Fig. 2f). Experiments at lower temperature (25 °C) indicate a related behavior with respect to the formation of transient DySSs, yet the lower activity of the BsaI enzyme leads to longer lifetimes and ca. 2000 bp higher $\overline{bp_w}$ in the DySS (Supplementary Note 1, Supplementary Fig. 1). Importantly, both enzymes are sufficiently stable in the investigated time regime to clearly assign the lifetime control to the ATP consumption (Supplementary Note 2, Supplementary Fig. 2).

**Transient sequence-definition and side group functionality in DySS oligomers and polymers (SfNAPs).** A striking advantage arising from the programmable nature of BsaI is the possibility to tailor sequences in multicomponent system, enabling a robust, modular and generic DNA toolbox to transiently arrange tailor-made DNA tiles in a predesigned order in a fuel-dependent fashion (Fig. 3a). This allows for complex transient systems design by using a library of DNA tiles (Fig. 3b). Desired outputs can be obtained by varying the combination of the inputs, resulting in sequence-defined oligomers and polymers. Figure 3c–e show versatile outputs by changing the input components for the multicomponent system. The respective combinations deliver major transient outputs up to the level of the tetramer ABCD, but for the pentamer ABCDE statistically expected oligomers ABCD and BCDE are also visible (Fig. 3e; Supplementary Note 3, Supplementary Fig. 3). Moreover, based on this principle, the modular DNA toolbox can further generate a range of transient alternating sequence-defined DNA polymers by properly programming the sticky ends on a binary DNA tile system (Supplementary Fig. 4a). Thus, this strategy opens pathways to transiently arrange functional groups along the DNA scaffold to introduce transient SfNAPs.

Next we address the challenge of introducing communicating near-space functionalities (intramolecular action) and pathways for side-chain multivalency (intermolecular action) by exploiting the sequence control in SfNAPs. Two important aspects need to be considered. First, ideally it would be possible to have strategies for arrangements of functions very closely together. This appears intuitively difficult as the cleavage/ligation sites have a space demand of at least 11 bp, which hampers a close positioning of units e.g. via nucleobase modifications in the DNA structure. We attempted the introduction of fluorophore and quencher in the N1-N5 region via non-native nucleobases, but this led to a failure of DNA cleavage (Supplementary Fig. 5). Secondly, one needs to consider—in a functional material context—that end group

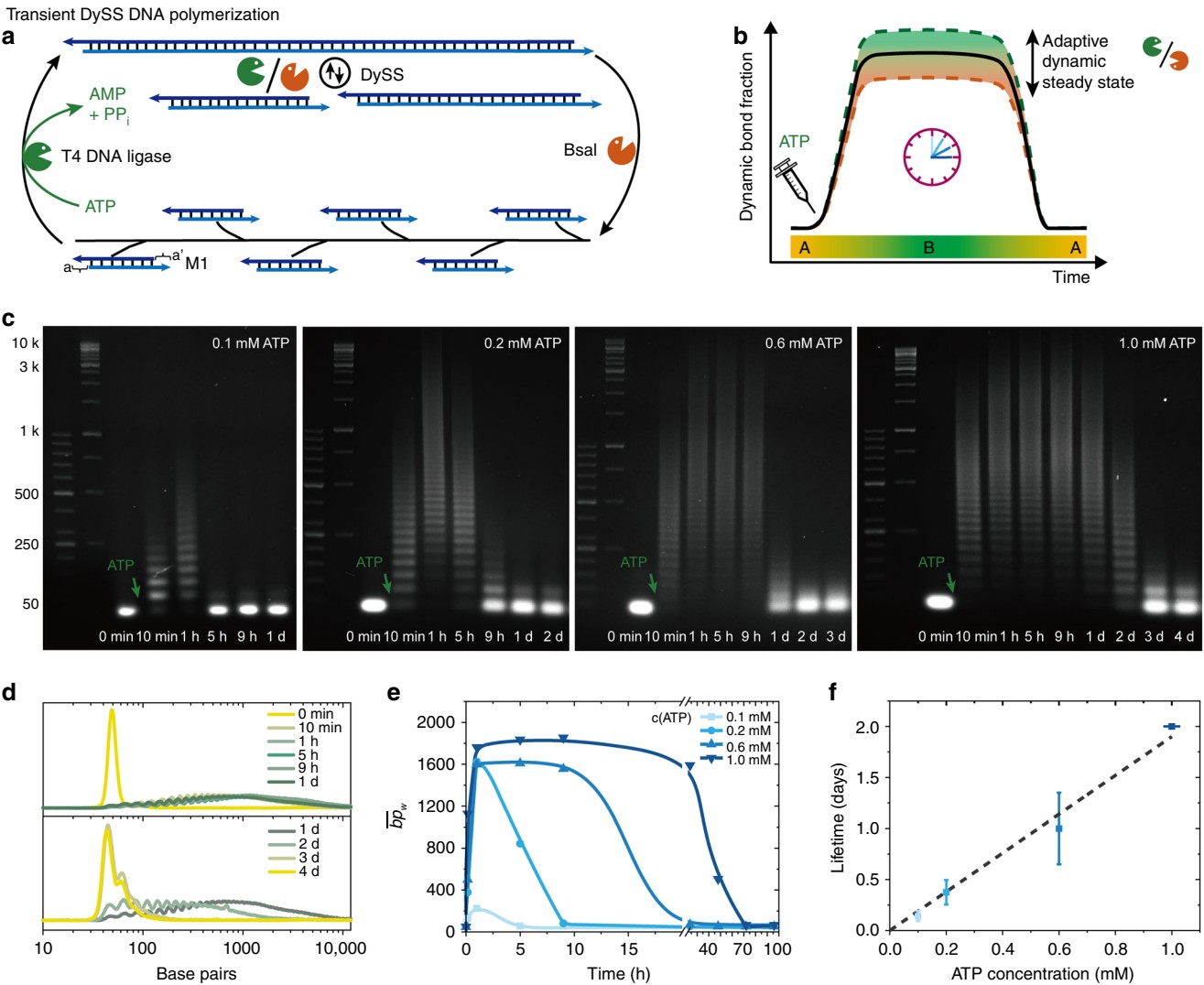

**Fig. 2 ATP-fueled transient DySS DNA polymerization. a, b** General scheme for ATP-powered transient DySS polymerization and control over lifetimes and adaptive DySS properties. The system enters into a DySS, state B, from its monomer state, state A, upon fueling with ATP, and returns to state A once the ATP is consumed. **c** Time-dependent AGE (2 wt. %, 90 V, 2 h) for transient DNA polymerizations with programmable lifetimes by fueling with varied ATP concentrations (0.1, 0.2, 0.6, and 1.0 mM). **d** Gray scale profiles from AGE at 1.0 mM ATP quantifying the transient shift of molecular weight, which is used to calculate the mass-weighted average chain length ($\overline{bp}_w$) for each kinetic aliquot. **e** The $\overline{bp}_w$ development with time by varying the ATP concentration from 0.1 to 1.0 mM. Lines are guides to the eye. Multiple ATP injections are shown in Fig. 3. **f** Lifetimes are controlled by the ATP concentration. Error bars are standard deviations of duplicate measurements. Conditions: 37 °C, 0.05 mM M1, 0.46 WU μL$^{-1}$ T4 DNA ligase, 2.5 U μL$^{-1}$ BsaI, and varying amounts of ATP.

modifications of ssDNA are preferred due to more facile synthesis compared to in-chain nucleobase modifications.

To achieve adjacent positioning of functions, we designed a refined multicomponent concept with very weakly binding tiles that only become consecutively activated for DySS polymerization upon covalent ligation. After a systematic investigation (Supplementary Note 4, Supplementary Fig. 4), we constructed systems using two dsDNA tiles, A and C, and 1 ssDNA tile, B (Supplementary Fig. 4f). Tile B is partially complementary to both tiles A and C, and it covalently connects with tile A via ligation to promote a further ligation between tiles A and C. Only one side of tile B is ligated, leaving adjacent non-ligating 3′ and 5′ available for modifications. The rest 4 nt sticky ends on tiles A and C form full junctions by ligation, rendering alternating DySS DNA polymers. Those are exclusively brought into close proximity upon DySS polymerization (Fig. 3f), because all three tiles are separated from each other in the absence of fuel due to

very weak base pairing at 37 °C (Supplementary Note 6, Supplementary Fig. 6). Tile A can either first react with tile B or tile C, followed by reacting with the third tile to show a FRET (Foerster Resonance Energy Transfer) in the polymers (Fig. 3g).

As direct optical read out function, we attached a fluorophore (Cy3) to the 5′ end of tile B and a quencher (Iowa Black FQ) to the 3′ end of tile C and monitored the formation of the proposed DySS structure with direct adjacent positioning of the functions via FRET (Fig. 3f). The transient FRET is visible after ATP injection. The ERN starts the dynamic polymerization with a decrease of the fluorescence intensity and the development of a transient plateau in the DySS, followed by recovery after ATP consumption (Fig. 3j; control without ATP in Supplementary Fig. 7c). The FRET lifetime is a function of the ATP concentration (Fig. 3j insert). Time-dependent AGE corroborates the presence of transient SfNAPs with similar lifetimes (Fig. 3h–j; Supplementary Fig. 7b, controls without modification in

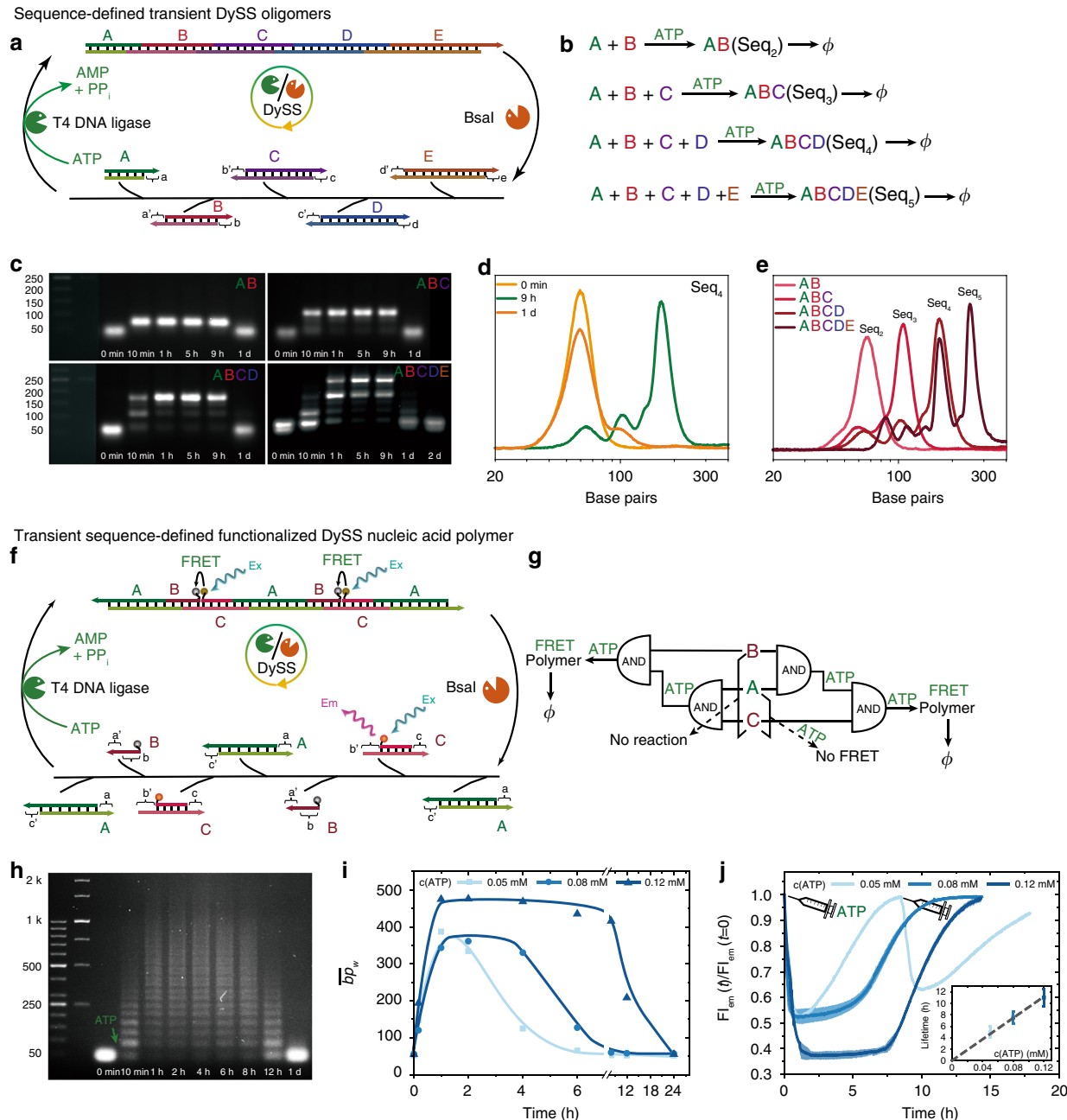

**Fig. 3 ATP-fueled transient sequence-defined functionalized nucleic acid oligomers and polymers (SfNAPs). a–e** Combinatorial organization of transient and programmable sequence-defined oligomers composed of varying inputs with orthogonally complementary ligation sites. *Phi* indicates disassembly after ATP consumption. **c** Time-dependent AGE with **d** extracted gray scale profiles for the transient oligomerization of the tetramer and **e** the distribution profiles of all oligomers in the DySS. **f–j** ATP-fueled DySS transient SfNAP with transient FRET signaling function. **g** Logic pathways for the ligation of the inputs to achieve FRET DNA polymers. **h** Time-dependent AGE (2 wt.%, 90 V, 2 h) of the transient SfNAP fueled with 0.12 mM ATP. **i** The $\overline{bp_w}$ development with time at different ATP concentrations. Lines are guides to the eye. **j** Transient FRET signaling at different ATP concentrations ($\lambda_{ex} = 530$ nm), and repeated fuel addition for 0.05 mM ATP; insert: FRET lifetime control by changing the ATP concentration. Shaded areas and error bars correspond to standard deviations from duplicate measurements. Conditions for **a**: 0.05 mM dsDNA tiles in total; 0.2 mM ATP, 0.92 WU μL⁻¹ T4 DNA ligase, 1.0 U μL⁻¹ BsaI, 37 °C. Conditions for **f–j**: 8.0 μM dsDNA-A, 8.0 μM dsDNA-C, 8.0 μM ssDNA-B, 0.92 WU μL⁻¹ T4 DNA ligase, 1.0 U μL⁻¹ BsaI, varied concentration of ATP, 37 °C.

Supplementary Fig. 7a), and confirms that also multicomponent tile systems can be efficiently driven into long DySS polymers. Higher ATP concentrations (0.08 and 0.12 mM) promote the $\overline{bp_w}$ from 380 bp to around 480 bp in a sustained DySS polymerization (Fig. 3i). Running the system with 1.2 equivalent amounts of B (Supplementary Fig. 7d) leads to a similar result as for 1.0 equivalent (Fig. 3h). The addition of a bit more tile B helps to diminish e.g. slight pipetting errors, ensuring efficient polymerizations.

This method connects fuel-driven DNA systems with SfNAPs, realizing autonomous fuel-driven organization of functional groups on a DNA scaffold. It shows advantages and different features over previous reports that use long template strands to guide the ligation of functional group-modified short ssDNAs

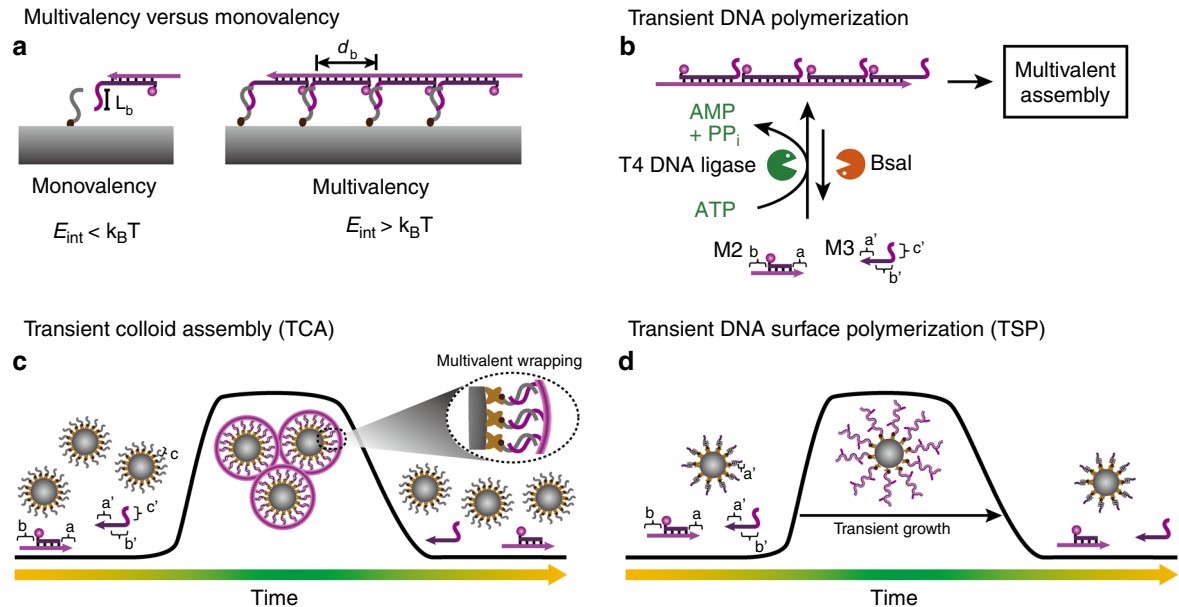

**Fig. 4 ATP-fueled hierarchical transduction of multicomponent SfNAPs to colloidal systems. a** Monovalent vs. multivalent binding with programmable interaction strength, $L_b$, and branch distance, $d_b$. **b** Schematic illustration of transient SfNAPs. **c** Schematic representation of the ATP-fueled transient DNA wrapping-induced TCA. **d** Schematic representation of ATP-fueled TSP on a colloidal surface by encoding a docking strand on the colloid (a') that integrates into the main chain of the DySS SfNAP.

along the template, by its template free, all in one system, and energy-dissipative, transient and dynamic manner[56,57]. For the active system, it is also very important that one can repeatedly fuel the system to obtain multiple transient signaling whenever there is chemical fuel (Fig. 3j; Supplementary Fig. 7e).

**Hierarchical transduction by transient cooperative multivalency.** The above strategy toward side group organization even in an adjacent manner on the transient DySS SfNAPs opens doors to translate the molecular fuel-driven polymerization systems to higher hierarchical levels, which we will demonstrate for the colloid assembly on micrometer length scale. Colloids are considered relevant model systems for complex systems and are of relevance for photonic or biomaterials applications[58,59]. Traditionally, thermo-reversible ssDNA interactions have been used to program the self-assembly of DNA-coated colloids, and strand displacement reactions, or direct enzymatic reactions have broadened the scope[60–65]. Here, we provide a strategy to control colloidal assemblies via our transient modular DNA toolbox and using ATP as molecular fuel.

The key concept is to realize that transient SfNAPs can be used to build up transient multivalent DNA interactions that only become cooperatively active in the autonomous driven DySS polymer state, while they are absent in the monomer state (Fig. 4a). To this end, we focus first on designing transient multivalency between SfNAPs with side-chain branches (c') and paramagnetic colloids (1 μm in diameter) coated with complementary ssDNA docking strands ($DoS_1$). Figure 4b shows the design of the DNA toolbox for transient multivalent binding. The DNA toolbox is changed to 1 dsDNA (M2) and 1 ssDNA tile (M3). M3 has three parts: A 4 nt overhang on its 5′ end is complementary to the 4 nt sticky end on M2 (a'/a). T4 DNA ligase can couple them. Another 8 nt part following the 4 nt overhang on M3 is complementary to the other sticky end on M2 (b'/b), but T4 DNA ligase cannot join these parts on this side of the DNA duplex, as the last part (c') of M3 is non-complementary to M1 and forms the dangling branch. A spacer of 5T is added before c' on M3 to increase the branch flexibility. Critically,

elongated SfNAPs can only form by ligation as the tiles otherwise remain disassembled due to the low $T_m$ of the overhangs. The interaction between each branch and the docking strand on the colloid (c'/c) needs to be designed in its length, ($L_b$, $T_m$), to be unstable at the system temperature of 37 °C (Fig. 4a). However, once ATP is added, all tiles (M2-M3) are dynamically polymerizing into transient DySS SfNAPs, then allowing for multivalent interactions that realize a cooperative bond formation as a function of branch density, $d_b$, and number of branches (length of the SfNAP). As a result, the DySS SfNAPs assemble onto the colloid surface, and more importantly, can bridge between various colloids due to their length and drive them into assembled structures, termed transient colloidal assembly (TCA, Fig. 4c). Alternatively, if the docking strand on the colloid is designed to integrate into the main chain of the SfNAP (a'/a) and does not feature complementary branch interactions, the SfNAPs transiently nucleate on the colloid surface and lead to a transient surface polymerization (TSP) and encapsulation without colloid assembly (Fig. 4d). Since the "driven multivalent glue" consumes the ATP to maintain its DySS, the effective lifetime of the TCA or TSP is coupled to the consumption kinetics of the molecular system.

Experimentally, we fixed the branch/docking strand interaction c'/c to CGAATAGA/TCTATTCG ($T_m = 33$ °C) and confirmed that dye-labeled multivalent SfNAPs with high density of branches ($d_b$ of 38 bp; P1–$d1_b$) yield efficient binding on the colloidal surface and consecutive assembly, while SfNAPs having a $d_b$ of 76 bp (P2–$d2_b$) and 117 (P3–$d3_b$) only lead to slight assembly or do not show assembly at all, respectively (Supplementary Note 8; Supplementary Fig. 8). Thus, a high branch density is needed for multivalent binding. Based on the above screening, we carried out the fuel-driven TCA transduced by transient SfNAPS with a $d_b$ of 38 bp. Confocal laser scanning microscopy (CLSM) visualizes the transient cluster formation, and strongly reduced single particle fractions can be quantified (Fig. 5a, b, f; AGE in Supplementary Fig. 8g). Corresponding AGE of the SfNAPs shows no significant difference between the transient DNA polymerization with and without colloid

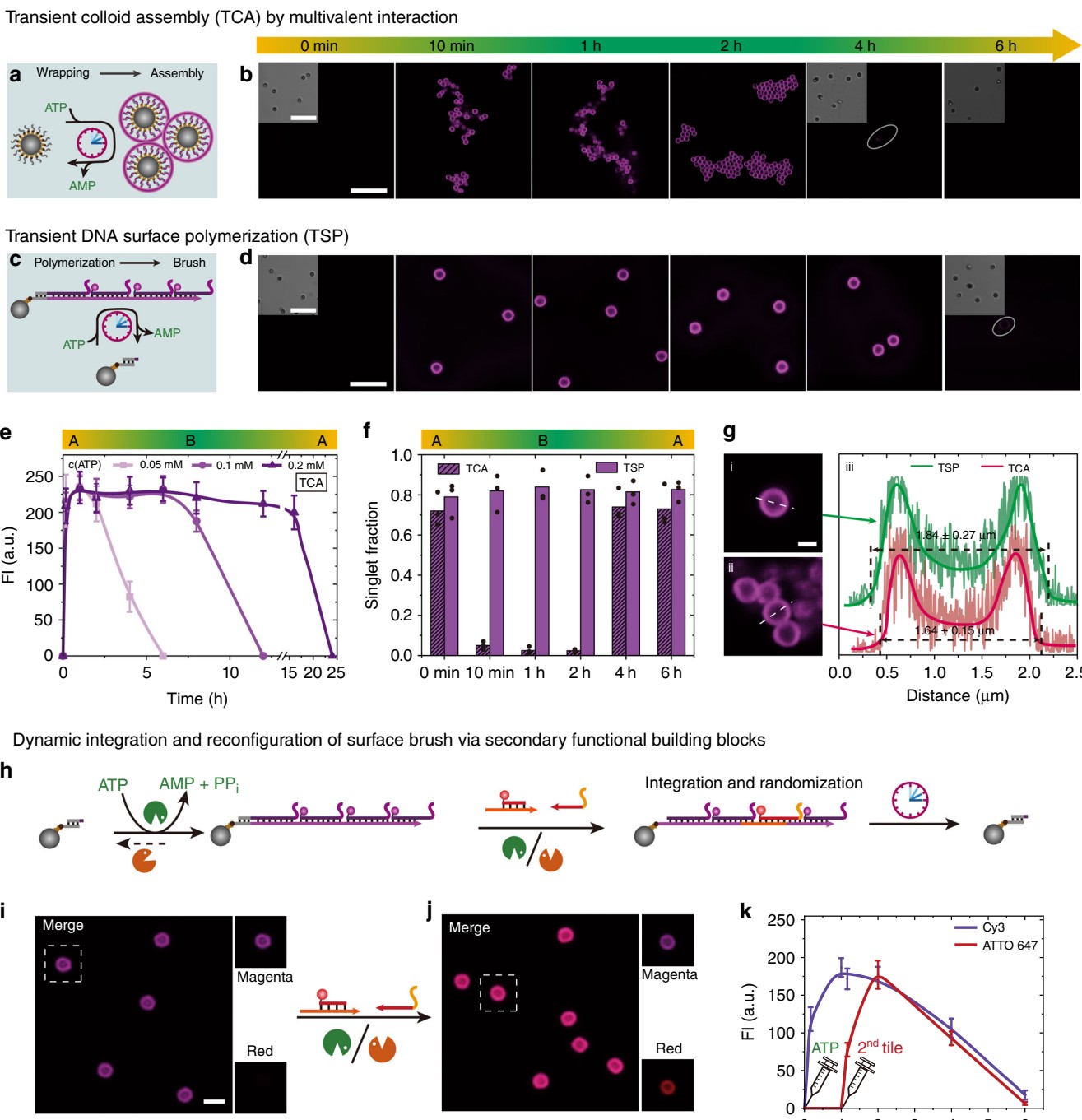

**Fig. 5 ATP-fueled transient colloid assembly (TCA), and dynamic DNA surface polymerization (TSP). a** Schematic illustration of DNA wrapping-induced TCA. **b** Time-dependent CLSM of TCA. Scale bars = 10 μm. **c** Schematic representation of TSP. **d** Time-dependent CLSM of transient DNA growth on colloidal surface. Scale bar = 5 μm; insert = 10 μm. **e** Time-dependent fluorescence intensity of the fluorescent shell for TCA with programmable lifetimes. Lines are guides to the eye. Error bars are standard deviations of 50 colloids. **f** Singlet fraction of the colloids for TCA ($d1_b$ = 38 bp) and TSP. Points correspond to analysis at random places. **g** Thickness of the fluorescent shells for TSP and TCA at 1 h (i: TSP, ii: TCA, iii: extracted spectra of fluorescence intensities from i and ii). Scale bar, 1 μm. **h** Secondary DNA tile integration and surface function reconfiguration during a running TSP. The second tile with ATTO 647 was added 1 h after the Cy3 tile. CLSM before (**i**) and after (**j**) adding the ATTO 647 tile confirm the integration and reconfiguration. Scale bar = 2 μm. **k** Time dependent fluorescence intensities on the colloid surface. Lines are guides to the eye. Error bars are standard deviations of 50 colloids. Conditions for **a**: 10.0 μM dsDNA-Cy3, 12.0 μM ssDNA, 0.92 WU μL$^{-1}$ T4 DNA ligase, 1.0 U μL$^{-1}$ BsaI, ca. 0.167 mg mL$^{-1}$ docking strand modified colloids (ca. 1.17–1.67 × 10$^8$ beads mL$^{-1}$), and 50 μM ATP, 37 °C. Conditions for **c**: same condition as **a** but using dsDNA functionalized colloids bearing sticky end a' on the docking strand to initiate a surface tethered main chain of the DySS SfNAP. All CLSM measurements were conducted consecutively with the same microscopy settings (see Supplementary Table 1 for sequences).

(Supplementary Fig. 8h). By varying the ATP concentration from 0.05 to 0.2 mM, the lifetime of the fluorescent shell, and thus also the TCA, can be programmed from 4 to 16 h (Fig. 5e; Supplementary Fig. 9). The fluorescent shell shows a thickness of ca. 320 nm (Fig. 5g, ii). Concurrently to the TCA, the SfNAP coating becomes visible as a transient fluorescent coating and clarifies the SfNAPs to be at the origin of the self-assembly (further control experiments with dye-labeled colloids in Supplementary Fig. 10).

Besides the transient multivalency programmed TCA, we also describe a method to control transient encapsulation and release of single particles via a DNA shell (TSP). To do so, we combined the same SfNAP in presence of colloids bearing a docking strand with an a' function able to integrate into the main chain of the SfNAP (Fig. 4d). This only allows for a TSP of freely dispersed colloids during the ATP-fueled transient polymerization (Fig. 5c, d, f). TSP leads to a thicker DySS fluorescent shell of ca. 420 nm compared to TCA (Fig. 5g, i; AGE in Supplementary Fig. 10e). After ATP consumption, the encapsulated particles are released. This demonstrates a strategy for the transient encapsulation of colloidal objects, and potentially even cells, with tunable transient coatings containing opportunities for dense functionalization.

To probe for actual dynamics in the DySS of the transient polymer graft layer, we added ATTO 647-labeled tiles (ATTO 647, $\lambda_{ex} = 638$ nm, $\lambda_{em} \approx 669$ nm, red) to an already active TSP system based on Cy3-labeled tiles ($\lambda_{ex} = 552$ nm, $\lambda_{em} \approx 570$ nm, magenta; Fig. 5h). Since the DySS SfNAP on the surface is in its dynamic covalent fueled state, the new tile integrates quickly across the full shell, as can be seen by dual color imaging and incorporation and enrichment of the ATTO 647-labeled tiles appearing in red (Fig. 5i, j). The integration proceeds within 10 min, which in turn indicates high dynamics despite a brush-like regime of the surface grown polymers (Fig. 5k; Supplementary Fig. 11). The fluorescence of both fluorophores on the shell disappears after ATP consumption confirming complete recovery of the original surface. In an application setting, this demonstrates that a surface functionalization can be adaptively reconfigured in the DySS, which would not be possibly in a static equilibrium system. The quick reconfiguration is a direct result of driving the structures here in a highly adaptive DySS with a flux-like character. This necessitates to synchronize energetic events with structural events, which is after all a decisive feature of the presented ATP-driven dynamic covalent bond system.

**Translating programmable ATP-fueled multivalent hierarchies into 4D systems behavior.** In the last part, we join the two enabling key aspects of this work, (i) ATP-dissipating programmable molecular recognition units and (ii) DySS SfNAPs with programmable multivalency to target hierarchical structures, into the development of the first out-of-equilibrium 4D transient self-sorting (TSeSo) model on a multicomponent systems level.

To this end, we designed two orthogonal transient DySSs SfNAP systems (Cy3-SfNAP-1, ATTO 647-SfNAP-2) with orthogonal sticky ends and branches and equipped two sets of colloids with appropriate docking strands (TSeSo-A, Fig. 6a). Before ATP addition, all colloids are freely dispersed and can only be visualized in bright field mode. The addition of ATP drives both SfNAPs into their separate DySSs, whereupon the built-up of multivalency patterns induces selective wrapping onto the respective colloids with the complementary docking strands, ultimately leading to transient narcissistic self-sorting in colloidal self-assemblies (TCA-1; TCA-2; Fig. 6c, f). Again the process is fully transient and clean disassembly of all colloidal aggregates occurs upon consumption of ATP. AGE of time-dependent aliquots visualizes transient nature of the SfNAPs regarding their length distribution (Fig. 6e). The wrapping and assembly of both SfNAPs shows similar kinetics, as the increase of the fluorescence intensities of the shells exhibit similar speeds (Fig. 6g, TSeSo-A), but, we suggest that different assembly time scales could be achieved by varying the sticky end overlaps.

By changing the docking strand on one colloid to a sequence that integrates into the main chain of one of the SfNAPs, a different transient self-sorting system (TSeSo-B) can be designed (Fig. 6b). In this system, the Cy3-SfNAPs tiles only polymerize on the colloid surface and lead to a transient encapsulation (TSP), while the unchanged ATTO 647-SfNAP tiles polymerize and then assemble their colloidal partners via multivalency into clusters (TCA-2; Fig. 6d). Interestingly, there is a slight aggregation of both colloids at very early stages of the ATP-fueled process (~10 min), whereafter however a clean separation into fully self-sorted encapsulated particles (Cy3-SfNAPs) and aggregated particles (ATTO 647-SfNAP) occurs. This means that some level of self-correction is operational due to the exchange dynamics during the DySS polymerization. Correspondingly, in the DySS plateau, the first set of particles show a high singlet fraction, while the second set shows a low singlet fraction/high assembly fraction (Fig. 6f). The fully autonomous behavior is visible by the transience of the encapsulation and assembly and the return to the original state of non-fluorescent particles after complete consumption of ATP. Overall, this demonstrates a first conceptual approach for hierarchically orchestrated, fuel-driven 4D self-sorting in multicomponent colloidal/DNA systems that are entirely programmed through chemically fueled molecular recognition events and in which emergent autonomous behavior, such as lifetimes, clusters, and surface ligand, display can be encoded.

## Discussion

Here, we have introduced the concept of fuel-driven programmable molecular recognition using ATP as fuel source and using DNA as versatile building blocks, which is enabled by ATP-powered ligation using T4 DNA ligase in balance with a class IIS restriction enzyme that has freely editable sticky ends. The lifetimes of the transient states are regulated by the ATP concentration, while exchange dynamics in the DySS is controlled by the enzyme concentrations. This concept hence steps away from other non-equilibrium DNA systems by operating in a template-free way and by not relying on transcription or polymerase machinery to make RNA or DNA strands that act downstream[38,40]. Our system enables significant flexibility as demonstrated for the transient arrangement of building blocks in predesigned sequences in oligomeric or polymeric structures (SfNAPs). Overall, the programmable nature of the dissipative and dynamic molecular recognition events gives rise to a highly modular DNA toolbox that enables multiple building blocks to run in parallel and orthogonally, which is an important stepping stone for reliable engineering of complex molecular multicomponent systems.

Higher levels of complexity in such systems can be achieved by attaching functional side groups inside the transient SfNAPs, which allows to introduce near space communication within a transient SfNAP or allows to implement the concept of fuel-driven cooperative multivalent binding between the SfNAPs and appropriately functionalized partners. We have established here that this transient multivalency can be tuned in its strength/ effectiveness to transiently cluster colloidal particles or to transiently encapsulate colloidal particles inside a functional shell. Due to its chemically driven and DySS nature, such an encapsulating shell can be adaptively reconfigured to dynamically incorporate new functional units upon addition of new building

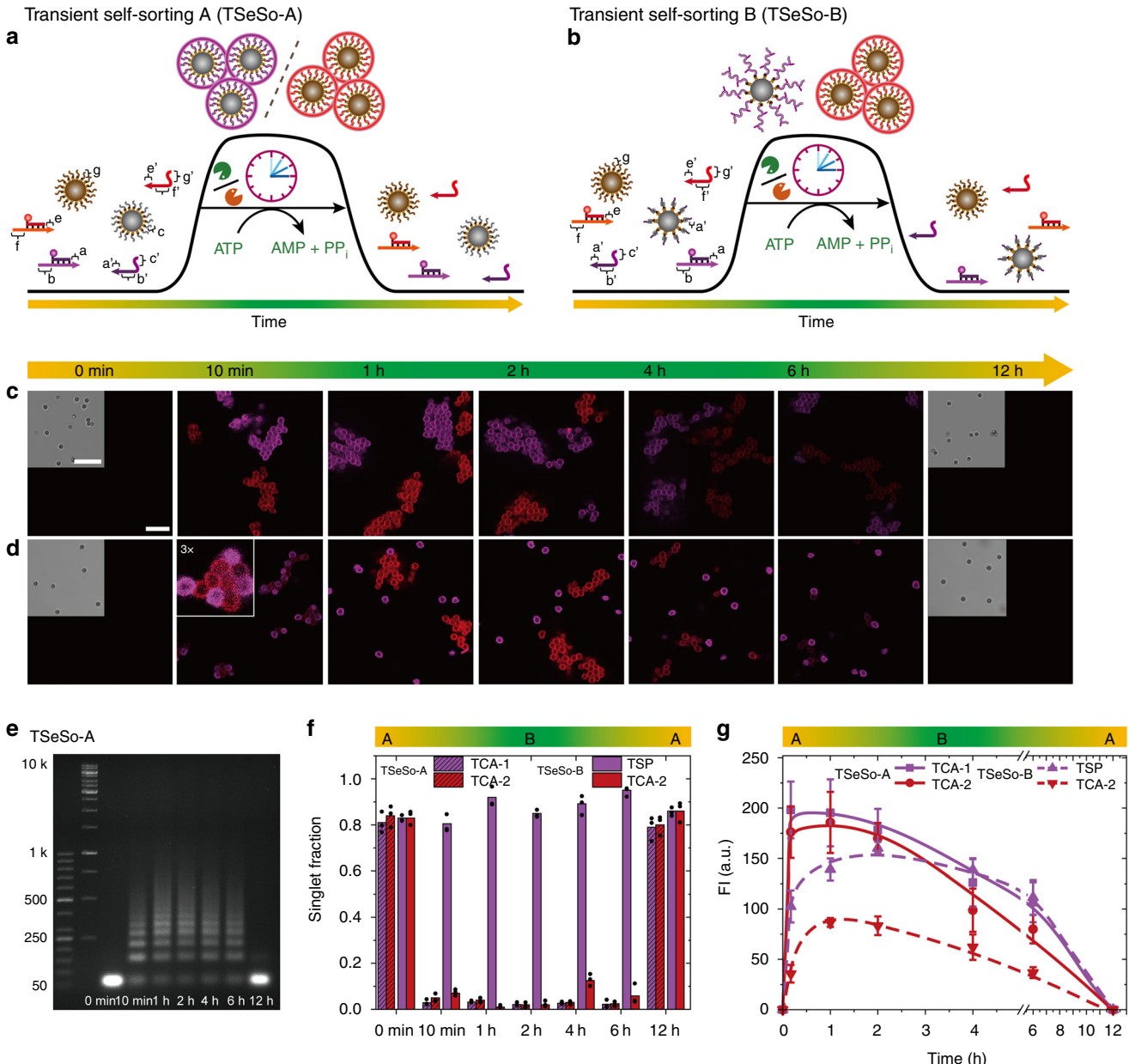

**Fig. 6 ATP-fueled transient self-sorting (TSeSo) on a multicomponent systems level. a, b** Schemes representing two different self-sorting multicomponent colloidal systems. **c** CLSM of two ATP-fueled colloidal assemblies, TSeSo-A, operating in the same systems in parallel. Scale bar = 5 μm; insert = 10 μm. **d** CLSM of TSeSo-B, whereby one colloid species is driven into assembly (TCA), while the other species is encapsulated (TSP). **e** Time-dependent AGE of the SfNAPs formed in TSeSo-A. **f** Time-dependent singlet fraction of both self-sorting systems. Points correspond to analysis at random places. **g** Fluorescence intensity of the colloids with time for both self-sorting systems. Lines are guides to the eye. Error bars are standard deviations of 50 colloids. Conditions for **a**: 10.0 μM dsDNA$_1$-Cy3, 12.0 μM ssDNA$_1$, 10.0 μM dsDNA$_2$-ATTO 647, 12.0 μM ssDNA$_2$, 0.92 WU μL$^{-1}$ T4 DNA ligase, 1.0 U μL$^{-1}$ BsaI, ca. 0.1 mg mL$^{-1}$ DoS$_1$ modified colloid, 0.1 mg mL$^{-1}$ DoS$_2$ modified colloid, 83.3 μM ATP, 37 °C. Conditions for **b**: The same condition as **a** except that the DoS$_1$ is changed to sticky-end a' terminated dsDNA. All CLSM measurements were conducted consecutively with the same microscopy settings (see Supplementary Table 1 for sequences).

blocks. The dynamics in such DySS SfNAPs may be important for the use in dynamic covalent libraries to target functions as in target recognition or molecular imprinting[57,66,67].

By combination of both concepts—ATP-powered and programmable molecular recognition with transient multivalency in SfNAPs—we further established transient self-sorting on a systems level with programmable lifetimes and showcased ATP-dissipating 4D self-sorting colloidal systems. This proof-of-concept demonstration highlights that the chemical energy inside a small molecular fuel can be transduced across four

hierarchical levels from simple dsDNA and ssDNA building blocks via SfNAPs to the formation of colloidal cluster and encapsulated colloids up to the systems level. Although this certainly falls short of the complexity of the fueled structure formation in the cytoskeleton, colloidal particles are important model systems and find applications for instance in photonics or for tissue engineering[58,59].

Looking out to the future, we envision that our strategy opens avenues to advance the fields of life-like materials, systems chemistry, and synthetic biology, as well as enables to better

understand some principles of living systems and provides means to engineer the interaction and communication of synthetic materials with biological systems. We foresee in particular that the concept of programmable transient multivalency will allow to study ATP-driven DySS liquid–liquid phase separation[68,69], control protein or cell assembly for programmable function and activity (e.g. tumor spheroid)[66,70], or to transiently regulate cell functions by multivalent ligand clustering to mediate signaling[71]. In the knowledge that ATP is the energy currency of living systems, it may also in long term provide a way for generating active matter in a cellular context. While we have so far engineered the multicomponent systems to operate in an orthogonal fashion, we also see possibilities that the interfacing of multiple dynamic systems can give rise to the formation of regulatory networks with more complex time-dependent behavior.

## Methods

**Hybridization of the dsDNA tiles**. All DNA strands were used as received (sequences of the DNA strands from $D_1$ to $D_{40}$ are in Supplementary Table 1), to which certain amounts of annealing buffer were added to make stock solutions and the concentrations were calculated via UV-VIS spectroscopy. The dsDNA tiles in this study were annealed from two complementary ssDNA with the same stoichiometry at room temperature overnight. The annealed stock solutions were stored at −20 °C for further use.

**ATP-fueled transient DNA polymerization**. Ligatable dsDNA tiles, M1, were first annealed from two complementary ssDNA, $D_1$ and $D_2$. The obtained dsDNA tiles carry two complementary sticky ends on the terminal sides and the BsaI recognition site on one side of the tile. The transient polymerization of such dsDNA tiles was carried out at 25 and 37 °C in 1x *NEB* CutSmart buffer (50 mM potassium acetate, 20 mM tris-acetate, 10 mM magnesium acetate, 100 µg mL⁻¹ BSA) with 0.05 mM M1, 0.46 WU µL⁻¹ T4 DNA ligase (HC, 20 WU µL⁻¹, Promega), and 2.5 U µL⁻¹ BsaI (20 U µL⁻¹, *NEB*). Different concentrations of ATP were added to fuel the system for programmable lifetimes of the assemblies. For analysis of the DNA polymer evolution with time, at different time intervals 6 µL aliquots of the reaction solution were quenched by 8 µL of the quenching buffer (200 mM EDTA, 10 mM Tris-HCl (pH 8.0), 50 mM NaCl). Afterwards, all quenched samples were analyzed by AGE (2 wt.% agarose gel in TAE buffer (40 mM Tris, 20 mM acetic ccid, 1 mM EDTA)) using 90 V for 2 h run time. The results were recorded by a transilluminator (UVsolo *touch* Gel electrophoresis documentation system, Analytik Jena), and the gray scales for the gel bands were further extracted by ImageJ to determine the distributions of the DNA polymers as a function of migrated distances. Standard DNA ladders were used to calibrate the intensity value distributions as a function of base pairs (bp), allowing to calculate the mass-weighted average chain length ($\overline{bp}_w$). All the AGE images were properly cropped to better display the data and save space[41].

**BsaI activity and stability investigation**. The enzyme activity with time was recorded by measuring the efficacy for the cleavage of dsDNA (annealed from $D_3$ and $D_4$) containing an inner restriction site by BsaI after different time periods of pre-incubation at 37 °C in CutSmart buffer. Briefly, several aliquots of BsaI were first pre-incubated in 1x *NEB* CutSmart buffer at 37 °C for 0, 12, 24, 48, and 72 h with an enzyme concentration of 1.5 U µL⁻¹ for each. Then, 0.025 mM dsDNA was separately added to the above pre-incubated restriction enzyme solutions. At different time intervals, 6 µL aliquots of the reaction solutions were quenched by 8 µL of the quenching buffer. Afterwards, AGE analysis (3 wt.% agarose gel in TAE buffer; 80 V; 2 h) was used to calculate the cleaved fraction. Results in Supplementary Fig. 2.

**ATP-driven sequence-defined DNA oligomer**. Five kinds of dsDNA tiles, A (annealed from $D_5$ and $D_6$), B ($D_7$ and $D_8$), C ($D_9$ and $D_{10}$), D ($D_{11}$ and $D_{12}$), and E ($D_{13}$ and $D_{14}$) were used to demonstrate the proof of concept for the ATP-driven transient sequence-defined DNA oligomers (Fig. 3a). The experiments with different tile combinations were run at 37 °C in 1x *NEB* CutSmart buffer with 0.05 mM dsDNA tiles in total (for each individual system, tiles are maintained in equal stoichiometry), 0.92 WU µL⁻¹ T4 DNA ligase, and 1.0 U µL⁻¹ BsaI. Then, the transient reaction of the sequence-defined DNA oligomer was initiated by 0.2 mM ATP. At different time intervals, 6 µL aliquots were removed, quenched and analyzed by AGE (3 wt.% agarose gel in TAE buffer; 80 V; for 2 h). Results in Fig. 3a–e, and Supplementary Fig. 3.

**ATP-driven alternating DNA polymers**. Two dsDNA monomers, A ($D_{15}$ and $D_{16}$) and B ($D_{17}$ and $D_{18}$) were used to make transiently alternating DNA copolymer (Supplementary Fig. 4a). The experiments were run at 37 °C in 1x *NEB* CutSmart buffer with 8.0 µM A, 8.0 µM B, 0.92 WU µL⁻¹ T4 DNA ligase, and 1.0

U µL⁻¹ BsaI and fueled by 0.24 mM ATP. To have free 5′ and 3′ on the DNA tiles for universal modifications, we modulated the system to one dsDNA, A ($D_{15}$ and $D_{16}$), and 3 ssDNA, B ($D_{19}$), C1 ($D_{20}$), and C2 ($D_{21}$), for the alternating DNA copolymer (Supplementary Fig. 4d). In this case, 8.0 µM of each dsDNA and ssDNA tiles were added to the system for transient DNA polymerization fueled by 0.24 mM ATP at 37 °C. To have fewer steps for achieving ligation to the alternating DNA copolymer, which may result in more efficient DNA chain growing, C1 and C2 were further elongated and pre-hybridized to have stable dsDNA C ($D_{18}$ and $D_{22}$) for the alternating DNA polymerization (Supplementary Fig. 4f). Thus, a new system with 2 dsDNA and 1 ssDNA with free 5′ and 3′ on the monomer for universal modifications was developed. Briefly, 8.0 µM A ($D_{15}$ and $D_{16}$), 8.0 µM B ($D_{19}$), and 8.0 µM C ($D_{18}$ and $D_{22}$) were added to 1x *NEB* CutSmart buffer supplemented with 0.92 WU µL⁻¹ T4 DNA ligase, 1.0 U µL⁻¹ BsaI. The transient DNA polymerization was initiated at 37 °C by adding 0.24 mM ATP. It is worth noting that, in this design, ssDNA-B should not play any stoichiometry function for the growth of the DNA copolymer. Therefore, the transient DNA polymerization was also carried by 1.2 equivalents of ssDNA-B without notable difference in product distribution. For all the above experiments, at different time intervals, 6 µL aliquots of the reaction solution were quenched by 8 µL of the quenching buffer, and analyzed by AGE (2 wt.% agarose gel in TAE buffer; 90 V; 2 h). Results in Supplementary Fig. 4.

**Cleavage of the dsDNA with modification to the nucleotide in the recognition and cleavage region of BsaI**. Fluorophore- and quencher-modified dsDNA tiles ($D_{23}$ and $D_{24}$) were dissolved in 1x *NEB* CutSmart buffer to make a concentration of 0.025 mM. Afterwards, BsaI was added to make a final concentration of 1.5 U µL⁻¹. Then, the solution was incubated at 37 °C for 4 h. Afterwards, samples at time zero and 4 h were analyzed by AGE (2 wt.% agarose gel in TAE buffer; 90 V, 1 h). Results are in Supplementary Fig. 5.

**ATP-driven transient SfNAP**. Using the above strategy towards alternating DNA copolymers with free 5′ and 3′, that end up in adjacent positions in the transient system, we attached fluorophores (Cy3) and quenchers (IABkFQ) to the free 5′ and 3′ of the DNA tiles for the alternating DNA copolymers aiming to achieve transient SfNAP with directly adjacent FRET pairs (Fig. 3f). Briefly, the transient SfNAP polymerization was carried out at 37 °C in 1x *NEB* CutSmart buffer with 8.0 µM A ($D_{15}$ and $D_{16}$), 8.0 µM C ($D_{18}$ and $D_{26}$), 8.0 µM B ($D_{25}$), 0.92 WU µL⁻¹ T4 DNA ligase, 1.0 U µL⁻¹ BsaI, and varied concentration of ATP. The transient SfNAP polymerization was fueled with 0.05, 0.08, and 0.12 mM ATP realizing programmable lifetimes for the SfNAP. The time-dependent fluorescence intensity of the system was recorded by fluorescence spectroscopy ($\lambda_{exc} = 530$ nm) to monitor the transient behavior. Samples were sealed to prevent evaporation. All experiments were simultaneously monitored by AGE (2 wt.% agarose gel in TAE buffer; 90 V; 2 h). Results are in Fig. 3h–j, and Supplementary Fig. 7.

**Repeated fueling of the transient SfNAP polymerization**. The experiments were carried out at 37 °C in 1x *NEB* CutSmart buffer with 8.0 µM A ($D_{15}$ and $D_{16}$), 8.0 µM C ($D_{18}$ and $D_{26}$), 8.0 µM B ($D_{25}$), 0.92 WU µL⁻¹ T4 DNA ligase, 1.0 U µL⁻¹ BsaI, and 0.05 mM ATP. After 500 min, another 0.05 mM ATP was added to the system to refuel the transient SfNAPs formation. The time-dependent fluorescence intensity of the system was monitor by fluorescence spectroscopy ($\lambda_{ex} = 530$ nm) to monitor the transient behavior. All experiments were simultaneously monitored by AGE (see above). Results are in Fig. 3j, and Supplementary Fig. 7e.

**Immobilization of the docking strand on streptavidin-coated, magnetic colloids (DoS₁)**. 10 µL of streptavidin-coated magnetic particles (Dynabeads™ MyOne™ Streptavidin C1, ThermoFisher Scientific) were washed with 10 µL CSF buffer (CutSmart buffer supplemented with 0.05% Pluronic® F-127) for three times. Afterwards, the particles were re-suspended in 20 µL CSF buffer with 0.05 mM DoS₁ ($D_{30}$) and the reaction was carried out at room temperature for 1 h. The particles were then washed by 10 µL CSF buffer 5 times, re-suspended in 10 µL CSF buffer, and stored in the refrigerator for further use.

**Static colloid assembly by branched SfNAPs with variable branch distance**. Three kinds of side chain-functionalized, branched SfNAPs with varied branch distances were first synthesized by ATP-powered ligation. For the DNA polymer with a branch distance of 38 bp, the experiments were run at 37 °C in 1× *NEB* CutSmart buffer with 10.0 µM dsDNA-Cy3 ($D_{27}$ and $D_{28}$), 12.0 µM ssDNA-branch ($D_{29}$), 0.92 WU µL⁻¹ T4 DNA ligase and 0.1 mM ATP for 12 h. To further increase the branch distance and investigate the effect of branch distance on multivalent binding, we set another spacer tile with variable length between the branched tiles (Supplementary Fig. 8a). Briefly, 10.0 µM Cy3-labeled dsDNA tiles annealed from $D_{18}$ and $D_{26}$, 10.0 µM spacer tile (dsDNA from $D_{16}$ or dsDNA from $D_{33}$ and $D_{34}$), 12.0 µM ssDNA-branch ($D_{29}$), and 0.92 WU µL⁻¹ T4 DNA ligase were dissolved in 1× *NEB* CutSmart buffer. Then, 0.2 mM ATP was added to start the polymerization and the system was kept running at 37 °C for 12 h. The SfNAPs were characterized by AGE (see above). Then, DoS₁-modified colloids were separately added to these three DNA polymers (ca. 0.167 mg mL⁻¹, ca. 1.17–1.67 × 10⁸ beads mL⁻¹). After 1 h, 3 µL of the suspension for each sample were extracted. The

particles were collected by a magnet and the supernatant was discarded. The collected microparticles were washed with 10 μL CSF buffer at 37 °C for two times and then re-suspended in 10 μL CSF buffer. The CSF buffer for washing was incubated at 37 °C in the thermoshaker. For each washing cycle, 10 μL CSF buffer was added to the microparticles, vortexed for 5 s, incubated in the thermoshaker for 2 min, and then the supernatant was removed via a magnet. Afterwards, the microparticles were characterized by CLSM. Results in Supplementary Fig. 8.

**Transient colloid assembly (TCA) by multivalent DNA wrapping**. The experiments were run at 37 °C in 1x *NEB* CutSmart buffer with 10.0 μM dsDNA-Cy3 ($D_{27}$ and $D_{28}$), 12.0 μM ssDNA-branch ($D_{29}$), 0.92 WU μL$^{-1}$ T4 DNA ligase, 1.0 U μL$^{-1}$ BsaI, ca. 0.167 mg mL$^{-1}$ DoS$_1$-modified colloids (ca. 1.17–1.67 × 10$^8$ beads mL$^{-1}$), and 0.05, 0.1, or 0.2 mM ATP. At different time intervals, 3 μL aliquots of the suspension were extracted. The particles were collected by a magnet and the supernatant was quenched by 2.5 μL quenching buffer. The collected microparticles were washed with 10 μL CSF buffer at 37 °C for two times and then re-suspended in 10 μL CSF buffer. The CSF buffer for washing was incubated at 37 °C in the thermoshaker. For each washing cycle, 10 μL CSF buffer was added to the microparticles, vortexed for 5 s, incubated in the thermoshaker for 2 min, and then the supernatant was removed via a magnet. Afterwards, the microparticles were characterized by CLSM. The quenched samples of the supernatant at different time intervals were collected and analyzed by AGE (see above). A reference experiment to compare the difference of the DNA polymerization with and without colloid was conducted and also analyzed via AGE. The experiment was performed as in the above protocol, except that the colloids were not added at the beginning. After ATP addition, the solution was immediately divided into two batches, and the colloids were only added to one batch of the solution. Afterwards, the chain length distribution was analyzed by AGE (see above). Results in Fig. 5 and Supplementary Fig. 8h.

**Transient DNA surface polymerization (TSP) from sticky-end terminated dsDNA functionalized colloids**. The colloids were first modified with dsDNA ($D_{31}$ and $D_{32}$) containing pendant, ligatable sticky ends on their terminus functionalization (similar to above). The surface polymerization was performed as for the TCA except that the sticky-end terminated dsDNA-modified colloids now served as initiators to anchor the transient DNA system along the main chains. The system was characterized by CLSM and AGE (see above). Results are in Fig. 5c.

**Secondary DNA tile integration and randomization on colloidal surface during TSP**. By utilizing the dynamic property of the ATP-driven transient ligation, we further investigated the dynamic integration and randomization of tiles to the dynamic DNA polymers grown on the colloidal surfaces (Fig. 5h). The experiments were run at 37 °C in 1x *NEB* CutSmart buffer with 10.0 μM dsDNA-Cy3 ($D_{27}$ and $D_{28}$), 12.0 μM ssDNA-branch ($D_{29}$), 0.92 WU μL$^{-1}$ T4 DNA ligase, 1.0 U μL$^{-1}$ BsaI, ca. 0.167 mg mL$^{-1}$ sticky-end dsDNA-modified colloids (ca. 1.17–1.67 × 10$^8$ beads mL$^{-1}$), and 50 μM ATP. After 1 h of TSP using tile 1 (Cy3-labeled dsDNA, annealed from $D_{27}$ and $D_{28}$), the second monomer (ATTO 647-labelled dsDNA; annealed from $D_{28}$ and $D_{38}$, 3 μM) together with 3.6 μM ssDNA-branch ($D_{29}$) were added to the system. Samples at different time intervals were collected, purified, and characterized with CLSM.

**TCA with an internal reference by colloid-immobilized fluorescent probes**. To have an internal reference and compare the fluorescent intensity development during TCA to an internally constant fluorescent signal, we elongated the DoS$_1$ (L-DoS$_1$; immobilized on the colloids) to feature another barcode close to the biotin motif and then hybridized another ATTO 488-functionalized ssDNA ($D_{35}$) to the L-DoS$_1$ ($D_{36}$), rendering the particles fluorescent without attachment of any transiently polymerizing DNA systems. The TCA in the presence of the ATTO 488-prefunctionalized colloids was carried out the same as normal transient TCA using ATTO 647-functionalized monomer systems (dsDNA annealed from $D_{28}$ and $D_{38}$). The results were also characterized by CLSM and AGE (see above). Results in Supplementary Fig. 10.

**ATP-driven transient self-sorting-A (TSeSo-A) of two TCA subsystems**. We further combined two orthogonal transient side-chain functionalized, branched SfNAP systems with two appropriately functionalized colloids in one reaction vessel to achieve self-sorting of the colloids. Subsystem 1 is composed of dsDNA$_1$-Cy3 ($D_{27}$ and $D_{28}$), ssDNA$_1$-DoS$_1$' ($D_{29}$, DoS$_1$' represents the sequence complementary to DoS$_1$ on the colloid), and DoS$_1$ ($D_{30}$)-functionalized colloids; subsystem 2 is composed of dsDNA$_2$-ATTO 647 ($D_{38}$ and $D_{39}$), ssDNA$_2$-DoS$_2$' ($D_{40}$), and DoS$_2$ ($D_{37}$)-functionalized colloids. Briefly, the experiments were run at 37 °C in 1× *NEB* CutSmart buffer with 10.0 μM dsDNA$_1$-Cy3 ($D_{27}$ and $D_{28}$), 12.0 μM ssDNA$_1$-DoS$_1$' ($D_{29}$), 10.0 μM dsDNA$_2$-ATTO 647 ($D_{38}$ and $D_{39}$), 12.0 μM ssDNA$_2$-DoS$_2$' ($D_{40}$), 0.92 WU μL$^{-1}$ T4 DNA ligase, 1.0 U μL$^{-1}$ BsaI, ca. 0.1 mg mL$^{-1}$ DoS$_1$ modified colloid, 0.1 mg mL$^{-1}$ DoS$_2$ modified colloid, and 83.3 μM ATP. The samples at different time intervals were collected and characterized by CLSM and AGE (see above). Results in Fig. 6.

**ATP-driven transient self-sorting-B (TSeSo-B) by combined TCA and TSP**. We further combined two orthogonal TCA and TSP systems. Subsystem 1 is composed of dsDNA$_1$-Cy3 ($D_{27}$ and $D_{28}$), ssDNA$_1$-DoS$_1$' ($D_{29}$), and colloids modified with dsDNA ($D_{31}$ and $D_{32}$) containing pendant, ligatable sticky ends on their terminus; subsystem 2 is the same as that in TSeSo-A. The experiments were run at 37 °C in 1× *NEB* CutSmart buffer with 10.0 μM dsDNA$_1$-Cy3 ($D_{27}$ and $D_{28}$), 12.0 μM ssDNA$_1$-DoS$_1$' ($D_{29}$), 10.0 μM dsDNA$_2$-ATTO 647($D_{38}$ and $D_{39}$), 12.0 μM ssDNA$_2$-DoS$_2$' ($D_{40}$), 0.92 WU μL$^{-1}$ T4 DNA ligase, 1.0 U μL$^{-1}$ BsaI, ca. 0.1 mg mL$^{-1}$ sticky-end terminated dsDNA ($D_{31}$ and $D_{32}$)-modified colloid, 0.1 mg mL$^{-1}$ DoS$_2$-modified colloid, and 83.3 μM ATP. The samples at different time intervals were collected and characterized by CLSM and AGE (see above). Results in Fig. 6.

## Data availability

The data that support the plots within this paper and other finding of this study are available from the corresponding author upon reasonable request.

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

## Acknowledgements
We acknowledge support by the European Research Council starting Grant (Time-ProSAMat) Agreement 677960, as well as from the Deutsche Forschungsgemeinschaft (DFG, German Research Foundation) under Germany's Excellence Strategy—EXC-2193/1–390951807 via "Living, Adaptive and Energy-Autonomous Materials Systems" (livMatS).

## Author contributions
J.D. and A.W. conceived the project. J.D. designed and performed all the experiments. A.W. designed experiments and supervised the project. J.D. and A.W. analyzed the data and wrote the paper.

## Competing interests
The authors declare no competing interests.
