## [Peer Review File · Nature Communications]

Reviewers' Comments:

Reviewer #1:

Remarks to the Author:

This reviewer is not confused; it is the authors that, by using imprecise terminology, send out a confusing message to the community. They fail to recognize the difference between different classes of fuel-driven systems. The system reported here relies on two subsequent covalent reactions (ATP-fuelled DNA-ligation followed by DNA-restriction: $A+ATP \rightarrow B+ADP \rightarrow A+P$, with the fortunate case that the final product is the same as the starting material (A). By continuously adding ATP the system can be maintained in a dynamic steady state with the intermediate compound B populated.

This is different from the examples 1-4 provided by the authors. In these examples, a chemical fuel is used to activate a building block altering its self-assembly properties. As a consequence, in all these systems two different energy dissipation pathways are present: building blocks can be deactivated in the monomeric state or in the assembled state. This is of importance, because the relative contributions of these pathways actually determines how chemical energy is exploited by the system. The differences have recently been discussed in a series of papers:

Ragazzon, Prins *Nat Nanotechnol.* 2018 Oct;13(10):882-889

Astumian *Nat.Communications.* 2019 10:3837

Penocchio, Rao, Esposito *Nat.Communications.* 2019 10:3865

Confusion arises because in this research area the same terminology is used to describe all systems without distinction. This is indeed confirmed by the discussion in the peer-review file of another paper in this area cited by the authors in their rebuttal letter (I was not that reviewer). Considering that the corresponding author is a main contributor in this area, it may be expected that he can distinguish the different scenarios.

Although I see why the authors have used the term out-of-equilibrium, I would suggest not to use it. Formally it is true that any ongoing chemical reaction (or sequence) is not at equilibrium, but it is not common practice in chemistry to describe chemical reactivity in these terms. What the authors have developed here is a cleverly designed way to maintain the system always moving towards the thermodynamic end state without ever reaching it. I think the terms dynamic kinetic steady state and transientness are very appropriate. I would further suggest to disconnect the word dissipative from structure (as in 'ATP-dissipative structures page 5' or '..to engineer dissipative structures' in the caption of Figure 1). The only structures in the system able to dissipate energy are the two enzymes.

Apart from this discussion on the used terminology, I think the work is of a high scientific level and offers a new perspective on DNA-nanotechnology. My previous comment on the incremental nature referred to the fact that the methodology of creating dynamic steady states had been published before. Here, the authors optimize that approach and exploit it. The obtained results are suitable for publication in *Nature Communications*.

Reviewer #3:

Remarks to the Author:

In this work, the authors extend their previously introduced concept to generate out-of-equilibrium dynamic steady states in DNA systems by combining the action of a ligase and a restriction endonuclease. In contrast to earlier work, the use of a type II-S restriction endonuclease allows to more freely choose the sequence involved (the same idea has been

previously used in DNA computing and is the foundation of the Golden Gate assembly method). With this new capability, they demonstrate the transient formation of DNA polymers, the transient assembly of colloidal particles, surface polymerization and also "self sorting".

This (new) reviewer thinks that after the revisions following the first round of reviewing the paper is generally acceptable for publication in Nat. Commun., but wishes to add a few notes.

As similar ideas have been around for a long time, this reviewer would disagree that the approach taken here is so radically new to justify statements like "break with these concepts and open a new avenue to 4D colloidal assemblies via our transient modular DNA toolbox and using ATP as molecular fuel." (and others)

- for instance, the author emphasizes the difference between the use of ATP as fuel and previous work using DNA/RNA polymerization. Still the systems are based on the same idea: one process "builds up" a structure, and a competing process degrades it. In fact, all the oscillating systems shown before can also show "transient behavior", when they are not driven in the right parameter range. It typically requires considerable fine-tuning to generate oscillations, while it is relatively easy to see the transient behavior also reported here – and this is done in part via the ratio of the enzymes added ...

Next to the work cited here (Ref. 38), the authors may also like to look at: E. Franco et al., PNAS 108, E784–93 (2011); J. Kim, Nucleic Acids Res. 42, 6078–6089 (2014).

The paper by Dehne et al. cited in 39 also shows transient colloid assembly – and the difference to the present work (of course, the authors emphasize ATP consumption etc.) is not that large ...

- The authors may be interested in the discussion on "fueling" computational processes *without ATP* using Type II-S enzymes in Y. Benenson, et al. PNAS, 100, 2191–2196 (2003).

- it is not clear to this reviewer why the authors use the statement "repurpose advanced Golden Gate cloning". Why the word "advanced"? It is not necessary to inflate statements all the time.

- the use of the word "tile" in this manuscript (here it is just used for double strands with sticky ends and one type of single strands) differs from the typical use of the term in DNA nanotechnology and may be confusing. Also "single-stranded tiles" such as those developed by Peng Yin's group are different as they are characterized by a very specific sequence domain structure.

Reviewer #1 (Remarks to the Author):

This reviewer is not confused; it is the authors that, by using imprecise terminology, send out a confusing message to the community. They fail to recognize the difference between different classes of fuel-driven systems. The system reported here relies on two subsequent covalent reactions (ATP-fuelled DNA-ligation followed by DNA-restriction: $A+ATP \rightarrow B+ADP \rightarrow A+P$, with the fortunate case that the final product is the same as the starting material (A). By continuously adding ATP the system can be maintained in a dynamic steady state with the intermediate compound B populated.

This is different from the examples 1-4 provided by the authors. In these examples, a chemical fuel is used to activate a building block altering its self-assembly properties. As a consequence, in all these systems two different energy dissipation pathways are present: building blocks can be deactivated in the monomeric state or in the assembled state. This is of importance, because the relative contributions of these pathways actually determines how chemical energy is exploited by the system. The differences have recently been discussed in a series of papers:

A) Ragazzon, Prins Nat Nanotechnol. 2018 Oct;13(10):882-889

B) Astumian Nat.Communications. 2019 10:3837

C) Penocchio, Rao, Esposito Nat.Communications. 2019 10:3865

Confusion arises because in this research area the same terminology is used to describe all systems without distinction. This is indeed confirmed by the discussion in the peer-review file of another paper in this area cited by the authors in their rebuttal letter (I was not that reviewer). Considering that the corresponding author is a main contributor in this area, it may be expected that he can distinguish the different scenarios.

Response: First of all we appreciate the comment in general. We do not fail to recognize the differences, but there is several points the reviewer does not consider: (1) The reviewer has not considered the concept of **transient multivalency**, which is the key part of this article and which then in fact leads to activated building blocks very much like the aforementioned studies (We denote this Structural Level 2 in the following). (2) The basic ATP-fueled reaction network is not a supramolecular polymerization as discussed in the aforementioned publications (A-C) (We denote this Structural Level 1 in the following level). It is in fact much closer to the well-researched topic of non-equilibrium steady state phosphorylation/dephosphorylation cycles (see below). (3) Along these lines, using a cyclic ABA system is not a “fortunate case”, but by design, and an important distinction to $A \rightarrow B \rightarrow C$ systems.

Hence the models described in the articles mentioned in A-C are in general directed to Structural Level 2, and in particular aim to quantify the “energy stored in a supramolecular self-assembly in a dissipative vs a thermodynamic state” as a measure for how much the self-assembled structure is out of equilibrium. Please see further comments below in particular on the relevance of the Structural Level 1. Hence things are more multifaceted and complex as the statement of the reviewer implies. As for terminology in general, AW has recently written a viewpoint in a different connection to clarify terminology in a similarly emerging field (Adv. Mater. 2019, 1905111). Emerging fields experience this “definition phase” and the reviewer is correct that many things are incorrectly defined in the past, at the moment, and become redefined or more properly defined as we move along and the field matures.

The question is whether this answer letter is the right place to discuss all of these issues, but we will make the case that the terminology in this article is correct.

General Fueled Self-Assembly Strategies in Literature

We start by looking at how energy is used in fueled systems in general with an emphasis on ATP:

Figure 1. Different implementation routines of ATP-driven dissipative self-assemblies with transient lifecycles.

Case (a): ATP is used as a co-assembling signal that is degraded over time by an external enzyme. Direct fuel to waste conversion takes place. There is no continuous cyclic reaction network. The environment does the job and there is no feedback system operating in the environment putting a reaction network in the environment outside equilibrium itself. The system shifts its assembly state in accordance with the equilibrium between building block (green), ATP/building block (orange), and co-assembly. Such a system can be realized with ATP-degrading enzymes. If ATP and ATP-degrading enzymes are used, the degradation of the ATP will most likely happen outside of the co-assembled structures (for kinetic reasons, diffusivity, access to the enzyme pocket etc.). Hence the process is a slow equilibration following Le Chatelier's principles. Such systems were already termed out-of-equilibrium (*Nat. Commun.* **6**, 7790 (2015); *Nat. Chem.* **8**, 725–731(2016)) even though the argument is rather that it is a slow equilibration followed from an energy rich state. The argument has been made to some extent by the same group, which introduced the case (a) systems, in the perspective in *Nat Nanotech.* **13**, 882-889 (2018) (article A). Perfectly fine, great systems.

Our system does not follow this pathway.

Case (b): A fuel-driven assembly according to the examples (1-4) and as discussed in the three papers A-C mentioned by the reviewer: A building block gets activated (e.g. hydrophobized) and thereafter assembles. Depending on the details of the kinetics (see papers A,B), this self-assembly (not talking about the entire system) can be in equilibrium or it can be out of equilibrium. For microtubules all deactivation happens in the structures and elaborate kinetic trapping occurs shifting it further away from equilibrium. Again if enzymes are involved in the deactivation pathway one can hypothesize that the free "activated" monomers are more accessible for a reaction with the enzyme. Depending on the K for the assembly, a part of those activated monomers takes part in self-assembly (e.g. supramolecular polymer). If k_{forward} and k_{backward} are fast, this is close to equilibrium. The models and theory discussed in the articles A-C mentioned by the

reviewer are intended for these systems that are dominated by supramolecular polymerization. They aim to quantify the “energy stored in the self-assembly in the dissipative vs the thermodynamic state” after hypothetically switching off the activation and deactivation pathways (is there any structural evolution yes/no). This serves as a measure for how much the self-assembled structure is out of equilibrium.

Our Structural Level 1 system does not follow this pathway, our Structural Level 2 however follows this pathway.

Case (c): The energy is directly used to build structures and the deactivation needs to occur on a structural level. There is no activated building block (except if one looks in the details of the activation pathway of the T4 ligase-mediated reaction; see SI of Sci Adv. 2019, 5 (7), eaaw0590). All energy goes directly into building the structure via covalent bonds, *that is of direct relevance to access the next length scale*. This is a profound difference to the dissipative system of case (b), where the chemical reaction and the energy transfer to make the activated building block only lead to an *activated building block of the same length scale* (a monomer for a supramolecular polymerization). In contrast to supramolecular systems, the structure formation in our system proceeds by covalent bonds and by installation of the energy-rich phosphor-ester bond ($\Delta G = -5.3$ kcal/mol). The deactivation is forced onto the structure. In dissipative supramolecular polymerizations (Case b) this bond is hardly considered, because it is of no relevance for a direct change of the structural length scale.

Cases (b) and (c) directly transfer the chemical energy to the system. *Importantly, none of the systems is similar to actin filaments, because the structures are not ATPases.*

Our Structural Level 1 system follows this case (c).

Along these lines, the reviewer mentions “... with the fortunate case that the final product is the same as the starting material (A).”. This is not a fortunate case, this is exactly the design principle and it is an important reaction network difference also in a thermodynamic sense (see below CQSS/NESS). Keeping the network in a cyclic fashion serves to generate different properties, in particular setting this strategy apart from RNA transcription and degradation networks which are $A \rightarrow B \rightarrow C$. This also leads to distinctly different use of energy in the systems, recycling of building blocks etc. Our system is not an $A \rightarrow B \rightarrow C$ of consecutive reactions, but is a reaction network with a “near perfect cycle” referring to structure build-up/destruction. Certainly, in any case, the backwards reactions are in principal possible for all reactions, but these rates are too small to be observed.

As a side note on details of our system, the ligation alone has in fact several consecutive substeps. The cutting, bond restriction (heat generation), is also not equivalent with immediate structure loss as the ssDNA overhangs generated are sticky (however only simulations of the enzymatic reaction could possibly shed light on the dynamics). Both aspects make the T4ligase/restriction enzyme system more complex than phosphorylation/dephosphorylation networks discussed below. The fueled DNA polymers carry the energy ($\Delta G = -5.3$ kcal/mol) in the structurally decisive bonds. The energy level of this is not an equilibrium ground state level and it is not an $A \rightarrow B \rightarrow C$ system, where the structure-forming entity would follow a downhill pathway complexity-type trajectory without regeneration. We believe that the following review Restrepo, D. Barragán and M. Rubi, Phys. Chem. Chem. Phys., 2019, DOI: 10.1039/C9CP01088B adds more general perspective on non-eq systems aside of supramolecular polymerizations and motors.

More details on the Structural Level 1 Network (fueled cyclic ABA networks): Such cyclic ABA systems with an energy-rich component are known extensively as non-equilibrium phosphorylation-dephosphorylation networks in theory and nature (see Figure, Hong Qian, Hao Ge, and others; e.g. Annu. Rev. Phys. Chem. 2007. 58:113–42; IEEE Transactions on NanoBioscience, 11, 3, 289, 2012,; Phys. Rev. E 87, 062125 (2013)). Therein the phosphorylated species is the non-

equilibrium species, and this is decoupled from the question of supramolecular self-assembly addressed in A-C. As pointed out in the paragraph just above, the T4 ligase/restriction system has more steps in covalent assembly and disassembly. Being precise, our system conforms to the closed quasi-steady state system (CQSS) because it does not have an outside, perfect ATP regeneration system (Phys. Rev. E 87, 062125 (2013)). Only if the system would have an ideal regeneration system with an unlimited regeneration capacity and perfectly similar diffusion rates between fuel and waste of the outside regeneration to the inside system it would be transformed into a perfect open non-equilibrium steady state system (NESS). CQSS and NESS have identical kinetics and positive entropy production rate, they distinguish themselves in e.g. heat housekeeping (Phys. Rev. E 87, 062125 (2013)). Importantly, a CQSS has a thermodynamic potential, it has a phosphorylation potential, ATP/ADP+P are not in equilibrium, the reaction cycle is driven, and equilibrium is only reached at the end. If equilibrium is reached at the end, everything before is not in equilibrium.

A more pressing question for the systems chemistry and non-equilibrium soft matter community: *Can we accept a non-equilibrium CQSS that moves towards equilibrium as a substantially appropriate mimic of a perfect NESS with a perfect regeneration.* We would make the argument, yes, under specific conditions; and if we want to learn something about systems design and behavior.

1. Rational: If the steady state in a CQSS is sufficiently stable over a certain time period of experimental observation, the changes of the concentrations of ATP/ADP+P are sufficiently miniscule so that they can be safely neglected for the behavior of the system at this time point. Additionally, fuel to waste conversion should not lead to substantial inhibition of the two chemical pathways (enzyme inhibition or anything else).

2: Experimental Reality: If we increase complexity of the structures formed and if we start using relatively precious building blocks (proteins, DNA, even complex supramolecular systems), we will be limited in the system size and how we can analyze such systems. Our colloid/DNA systems are maintained in well plates with glass bottom plates for microscopy observations with small μL volumes (40 – 100 μL), they cannot – under reasonable experimental constraints – be hooked up to a perfect ATP recycling system to adhere to perfect open systems conditions. Such systems cannot be operated in a continuously stirred tank reactor where only a spectroscopic readout would be possible, or in a dialysis bag, where time-dependent in-situ microscopy analytics are turning very difficult. Our systems show sufficiently long steady states.

3. What is the alternative? What is an ideal open system, what is an ideal regeneration system? If we want to perfectly conform to theory of thermodynamics in a NESS in a chemically fueled ABA cycle, we need to install a perfect or ideal regeneration system. On paper this can be done. In an experiment, we need to ask the question “where do we close the universe?” or “what is the realistic and needed universe”. *Option 1:* Certainly, we could have installed an enzymatic ATP regeneration system into our CQSS to regenerate ATP (e.g. using creatine phosphate and kinases) and make the system “more open” and “more steady state”. However, this is still not an ideal regeneration system, it just shifts the whole problem upstream. At some point the regeneration system will run out of fuel and we are back with the original problem that the system is in fact not fully open as with an ideal regeneration system. Also, a general waste problem may or may not arise. *Option 2:* Of course, systems could be placed in a dialysis reactor with influx of ATP and outflux of waste (if there is sufficient material available). This was done elegantly in (*Nat. Commun.* **8**, 15899 (2017)), but borne out of the necessity to overcome critical waste inhibition in their enzymatic reaction network system. But great to make the effort. Again if we want to be precise and conform 100% to NESS theory, we need to raise the question on the size of the outside supply (can be safely neglected in experiments), and how do we assure similar diffusion rates of fuel and waste so that there are no concentration changes in the supposedly steady state? Probably at best this can be dealt with a continuously stirred tank reactor (CSTR) and sufficiently large supplies, but this setup is just inapplicable for some studies involving increasingly precious materials that form complex and sensitive structures. Additionally, CSTR systems (no matter whether using a dialysis bag or a large reactor) need to be stirred to minimize concentration gradients. How do we deal with shear-responsive assemblies that behave differently in a stirred environment as compared to non-stirred environment? Stirring in 40 – 100 μL is hardly an option.

Such activation/deactivation networks in a CQSS setting are common in a range of “dissipative systems”. But, there is a difference for this dynamic covalent polymerization system and other supramolecular polymerization systems. Our system uses the reaction cycle to directly build up larger structures, while supramolecular polymerization systems only make activated monomers that then undergo structure formation. Hence, the non-equilibrium nature of the species B in our ABA networks is of much higher relevance for structure formation on a larger length scale compared to supramolecular polymerization systems.

More details on Structural Level 2: More importantly, for this article - and unfortunately making the whole discussion more multifaceted - the reviewer aims to make a difference between references 1-4 and this article. However, as we introduce the concept of **transient multivalency** here, the whole system essentially now turns our system from case (c) to the situation of examples 1-4 (hence case (b)). In addition to the basic reaction network of ligation and cutting (Structural Level 1), transient multivalency is used to make structures on an even higher hierarchical level (Structural Level 2). The SfNAPs are activated to become the assemblers, and the situation is similar to examples 1-4, except for the fact that distinct hierarchical structural layers are involved (small DNA; intermediate SfNAP; large colloidal assembly). This has been clearly shown in the central scheme of the article and indicated with the hierarchical levels. Since we are using multivalency and colloidal assembly it is of course not possible to try to identify dynamic instabilities as in fibrillar assemblies. Kinetic trapping is however evident in our assemblies on the colloidal domain, so these structures do not equilibrate locally as discussed above for specific situations of case (b).

Put simply, transient multivalency (and assembly) is conceptually the same as hydrophobization by methylation (and assembly), addition of a phosphate group to trigger electrolyte complexation (and assembly), etc.

As a summary, the contributions of each “reaction layer” need to be considered, and it is probably better to talk about non-equilibrium structure formation to also consider the covalent Structural 1 Level, and not only reduce the system to the supramolecular Structural level 2. Certainly, the Structural Level 1 is of low relevance for supramolecular polymerizations investigated by others before, and the question on energy storage in a supramolecular polymerization system is more relevant to define the non-equilibrium nature in such a system.

In general, for targeting or judging the non-equilibrium nature of systems one needs to consider more aspects to formulate a global picture: Network Level ($A \rightarrow B \rightarrow C$ downhill, vs $A \rightarrow B \rightarrow A$ cyclic, closed system vs. open system (how open is the system?)); any additional feedback or regulatory loops present?; Relevance of a non-equilibrium species B in an ABA network for structure formation on higher length scale of significance to the system?; How does any higher level assembly store energy? Related to this is the more fundamental question, whether a non-equilibrium CQSS migrating to equilibrium can fulfill the necessary criteria to mimic essential traits of a perfect NESS sufficiently, and under which experimental constraints one needs to operate to realize this. Certainly, behavioral richness increases with the “openness” of the system, and with the complexity of the regulatory network (BZ reaction or PEN toolbox show rich behavior in closed systems). Looking at recent literature from a range of groups, one cannot argue that we use the term in a substantially more light-minded way. Additionally, the reviewer has based the comment on not considering the transient multivalency as transition from case c to case b with presence of activated building blocks for higher level assembly.

We have added a related clarification of QSS/NESS in our revised manuscript, and also pointed in the caption of Figure explicitly to the two hierarchical levels and their structure formation. We also went carefully through the text on the point of the use of non-equilibrium.

Although I see why the authors have used the term out-of-equilibrium, I would suggest not to use it. Formally it is true that any ongoing chemical reaction (or sequence) is not at equilibrium, but it is not common practice in chemistry to describe chemical reactivity in these terms. What the authors have developed here is a cleverly designed way to maintain the system always moving towards the thermodynamic end state without ever reaching it. I think the terms dynamic kinetic steady state and transientness are very appropriate.

Response: As we discussed above and in the previous letter, this is not an $A \rightarrow B \rightarrow C$ reaction system, this is a chemical

reaction cycle with different levels of hierarchy. We have expanded the answer substantially now above. It passes along case c and case b as discussed above. The system clearly channels the energy through the system on two different hierarchical levels of structure forming entities. It is NOT an ATP co-assembly system with concurrent ATP degradation (case a). The non-equilibrium nature is clear from two aspects, the ABA bond network itself, the high-energy structures formed and the transient multivalency that forms structures even on higher length scales.

The reviewer also mentions “maintain the system always moving towards the thermodynamic end state without ever reaching it”. Not reaching it means not being in equilibrium. At the end, the final state is reached once the ATP is used up. All transient systems or chemical reaction networks working in closed system move towards equilibrium ultimately (also an “open” system with limited regeneration capacity), and they distinguish themselves in the use of the energy in the system (Figure 1). We would argue that the term “dynamic kinetic steady state” as suggested by the reviewer would make terminology unclear. Certainly, controlling the kinetics is key.

Terminology-wise, we should restrict the term far-from-equilibrium to systems undergoing bifurcation and showing e.g. oscillations or bistability. If the network is right, this does also not require an open system (BZ reaction, DNA PEN toolbox etc); Hence a closed system therein is accepted, but of course there is no doubt that such networks have high levels of complexity and feedback. In terms of the system approach one also needs to raise the general question: If the reaction network is a non-equilibrium feedback-controlled network (e.g. a BZ reaction in closed stirred tank reactor) and a self-assembly is coupled to this following perfect equilibration due to high dynamics, how do we term this? Is it an equilibrium self-assembly in a non-equilibrium environment? Is it a non-equilibrium self-assembling system? Probably it should not be termed a non-equilibrium self-assembly. I do not see a consensus on this at this point in time.

I would further suggest to disconnect the word dissipative from structure (as in ‘ATP-dissipative structures page 5’ or ‘.to engineer dissipative structures’ in the caption of Figure 1). The only structures in the system able to dissipate energy are the two enzymes.

Response: We understand what the reviewer wants to say and agree that the term should be more precise. The reviewer is correct that the structure is not an ATPase like actin. This is not the case for most dissipative self-assemblies. For this case, however, the enzymes are not ATPases like potato apyrase (also not in combination). The T4 ligase does not consume or convert ATP unless the DNA is there. The DNA in its design with the phosphorylation is mandatory to “dissipate the ATP” in the sense that it is consumed and used. We have changed to ATP-fueled structures to avoid this confusion.

Apart from this discussion on the used terminology, I think the work is of a high scientific level and offers a new perspective on DNA-nanotechnology. My previous comment on the incremental nature referred to the fact that the methodology of creating dynamic steady states had been published before. Here, the authors optimize that approach and exploit it. The obtained results are suitable for publication in Nature Communications.

Response: The dynamic steady states are only exploited here for some subparts in this manuscript. This publication is in fact about introducing the programmability in multicomponent systems, the transient multivalency to reach higher length scales and the self-sorting phenomena on a systems level.

Reviewer #3 (Remarks to the Author):

In this work, the authors extend their previously introduced concept to generate out-of-equilibrium dynamic steady states in DNA systems by combining the action of a ligase and a restriction endonuclease. In contrast to earlier work, the use of a type II-S restriction endonuclease allows to more freely choose the sequence involved (the same idea has been previously used in DNA computing and is the foundation of the Golden Gate assembly method). With this new capability, they demonstrate the transient formation of DNA polymers, the transient assembly of colloidal particles, surface polymerization and also “self sorting”.

This (new) reviewer thinks that after the revisions following the first round of reviewing the paper is generally acceptable for publication in Nat. Commun., but wishes to add a few notes.

As similar ideas have been around for a long time, this reviewer would disagree that the approach taken here is so radically new to justify statements like “break with these concepts and open a new avenue to 4D colloidal assemblies via our transient modular DNA toolbox and using ATP as molecular fuel.” (and others)

Response: We do not think that concepts like transient multivalency, chemically fueled molecular recognition and self-sorting have been around for a long time, or have been around at all. Basic activation/deactivation networks yes, but it really takes a different thinking to realize how this could be exploited for hierarchical systems design up to the level of self-sorting. We changed the sentence to “we provide a new strategy to control colloidal assemblies via our transient modular DNA toolbox and using ATP as molecular fuel.”

- for instance, the author emphasizes the difference between the use of ATP as fuel and previous work using DNA/RNA polymerization. Still the systems are based on the same idea: one process “builds up” a structure, and a competing process degrades it. In fact, all the oscillating systems shown before can also show “transient behavior”, when they are not driven in the right parameter range. It typically requires considerable fine-tuning to generate oscillations, while it is relatively easy to see the transient behavior also reported here – and this is done in part via the ratio of the enzymes added ...

Response: We agree that the basic principle behind making a transient system is the same: one process to build up a structure and a competing process to degrade it. As emphasized above there is a difference in ABC systems (RNA translation and degradation) and an ABA system (here). This also has implications for dynamics in a steady state and for structural design in general (e.g. transient multivalency) where different approaches offer different possibilities.

Next to the work cited here (Ref. 38), the authors may also like to look at: E. Franco et al., PNAS 108, E784–93 (2011); J. Kim, Nucleic Acids Res. 42, 6078–6089 (2014).

Response: we were aware of these papers. Both papers reported the same transcriptional oscillator system we cited in ref. 38.

The paper by Dehne et al. cited in 39 also shows transient colloid assembly – and the difference to the present work (of course, the authors emphasize ATP consumption etc.) is not that large ...

Response: The present MS differs significantly in concept and novelty of the approach: Ref 39 utilized transient RNA transcription to guide colloid clustering, in which the methodology was well established before. In our case, we establish the programming system from scratch, and use of ATP-fueled transient molecular recognition and transient multivalency for assembly. Additionally, we report self-sorting.

- The authors may be interested in the discussion on “fueling” computational processes *without ATP* using Type II-S enzymes in Y. Benenson, et al. PNAS, 100, 2191–2196 (2003).

Response: Thanks for the suggestion. We know the article, very nice work. We believe that the reviewer is making a general comment for this research direction.

- it is not clear to this reviewer why the authors use the statement “repurpose advanced Golden Gate cloning”. Why the word “advanced”? It is not necessary to inflate statements all the time.

Response: “advanced” removed. Advanced was used in contrast to normal cloning techniques using class I restriction enzymes.

- the use of the word “tile” in this manuscript (here it is just used for double strands with sticky ends and one type of single strands) differs from the typical use of the term in DNA nanotechnology and may be confusing. Also “single-

stranded tiles” such as those developed by Peng Yin’s group are different as they are characterized by a very specific sequence domain structure.

Response: The word “tile” has been used in many papers and in different research directions. People name lots of building blocks as a tile, such as DNA origami (Yan. *Nano letters*, 11(7), 2997-3002 (2011); Willner. *Nano letters*, 18(4), 2718-2724 (2018)), DNA nanostar (Yan. *ChemPhysChem*, 7(8), 1641-1647 (2006)), and as the reviewer suggested, ssDNA in Yin’s work.

Since we use a class II restriction enzyme the recognition and programmability of the sticky end is of real importance, which is why we believe that the use of the word tile is appropriate.